# Hetero-Deformation Induced Hardening in a CoCrFeNiMn High-Entropy Alloy

**Hamed Shahmir [1,\*], Parham Saeedpour [1], Mohammad Sajad Mehranpour [2], Seyed Amir Arsalan Shams [3] and Chong Soo Lee [3]**

1 Department of Materials Engineering, Tarbiat Modares University, Tehran 14115-143, Iran
2 School of Metallurgy and Materials, College of Engineering, University of Tehran, Tehran 11155-4563, Iran
3 Graduate Institute of Ferrous Technology, Pohang University of Science and Technology, Pohang 37673, Republic of Korea
\* Correspondence: shahmir@modares.ac.ir; Tel.: +98-2182883305

**Abstract:** One of the most important issues in materials science is to overcome the strength–ductility trade-off in engineering alloys. The formation of heterogeneous and complex microstructures is a useful approach to achieving this purpose. In this investigation, a CoCrFeNiMn high-entropy alloy was processed via cold rolling followed by post-deformation annealing over a temperature range of 650–750 °C, which led to a wide range of grain sizes. Annealing at 650 °C led to the formation of a heterogeneous structure containing recrystallized areas with ultrafine and fine grains and non-recrystallized areas with an average size of ~75 μm. The processed material showed strength–ductility synergy with very high strengths of over ~1 GPa and uniform elongations of over 12%. Different deformation mechanisms such as dislocation slip, deformation twinning and hetero-deformation-induced hardening were responsible for achieving this mechanical property. Increasing the annealing temperature up to 700 °C facilitated the acquisition of bimodal grain size distributions of ~1.5 and ~6 μm, and the heterogeneous structure was eliminated via annealing at higher temperatures, which led to a significant decrease in strength.

**Keywords:** hetero-deformation induced hardening; heterogeneous microstructure; bimodal grain size; microstructure engineering; high-entropy alloy





## 1. Introduction

The CoCrFeNiMn high-entropy alloy (HEA), known as the Cantor alloy, with a face-centered cubic (*fcc*) single phase structure, has gained attention due to its high ductility, cryogenic fracture toughness and corrosion resistance [1–5]. However, the strength–ductility trade-off of this HEA is an important and challenging dilemma, similar to other conventional alloys. The low yield stress and ultimate tensile strength of this alloy in fully annealed conditions, which are 300 and 530 MPa, respectively [6,7], have led to many attempts to improve the alloy's strength via grain refinement using severe plastic deformation or thermomechanical treatment, together with precipitation hardening [8–16].

Thermomechanical treatment, including cold rolling, to activate strain hardening followed by post-deformation annealing to obtain a favorable microstructure is a process that is easy to develop for industrial applications. However, the degree of plastic deformation, ranging from small to heavy/severe plastic deformation, the processing temperature (cryo- to warm rolling), annealing temperature (500–1000 °C) and time (a few to a hundred minutes) are control parameters for microstructure engineering, used to obtain desirable mechanical properties [17,18]. The alloy strength increased slightly after applying the thermomechanical procedure due to the formation of fine or ultrafine grains. The fabrication of nanograin structures using high-pressure torsion led to a significant increase in the strength (>1.5 GPa) at the expense of a dramatic decrease in the ductility (elongation of <4%) [19]. Post-deformation annealing at an appropriate temperature for

an appropriate time was used successfully to restore the ductility without significantly decreasing the strength [19–21]. For example, post-deformation annealing at 750 °C for 60 min led to a yield stress of ~1 GPa and elongation of 20% [21]. The annealing of the alloy is a challenging procedure because the heat treatment of the alloy at 200–700 °C leads to decomposition and the formation of various precipitates, including MnNi-rich *fcc*, FeCo-B2, Cr-rich *bcc* and Cr-rich sigma, especially after deformation, which significantly affects the mechanical properties of the alloy. In this case, it was reported earlier in different investigations that the formation of brittle sigma precipitates deteriorates the ductility [19–22]. Most investigations performed earlier were focused on full recrystallization and tailoring the microstructure through grain refinement, together with appropriate precipitates, to achieve optimal mechanical properties [18–23]. Severe or heavy deformation may promote precipitation due to the existence of a large number of grain boundaries and dislocations as fast diffusion pathways and preferential nucleation sites during short-term annealing at intermediate temperatures [21].

Recently, a new approach was developed to improve the mechanical properties through hetero-deformation induced strengthening (HDI) and strain hardening by creating a heterogeneous microstructure using a specific thermomechanical treatment [24]. In this structure, the dislocation hardening of pre-existing dislocations in the deformed area and the formation of back stress due to the generation and accumulation of geometrically necessary dislocations (GNDs) in the soft recrystallized area and near the area boundaries are two important phenomena that occur during straining [24–26]. Plastic deformation through cold rolling followed by partial recrystallization during post-deformation annealing at intermediate temperatures is a simple method for achieving the heterogeneous structure. The structure is decorated with recrystallized and non-recrystallized regions with heterogeneity characteristics such as a given range of grain size, pre-existing heterogeneous dislocations, retained deformation twins and precipitates in the HEAs, which may offer superior mechanical properties compared with the fully recrystallized material [27–31]. It is important to note that the suppression of brittle sigma phase formation is a key factor required to retain ductility during partial annealing.

Although cold rolling followed by partial recrystallization annealing is an easy process to apply, it is difficult to control the microstructural evolution. Furthermore, the appropriate annealing process for achieving a precipitate-free heterogeneous microstructure in the equiatomic CoCrFeNiMn HEA after medium plastic deformation has rarely been investigated. The present investigation focused on a strategy based on thermomechanical treatment, including medium plastic deformation applied via cold rolling followed by post-deformation annealing at elevated temperatures to facilitate partial recrystallization, with no formation of brittle precipitates. The study of the hetero-deformation-induced strengthening and strain-hardening mechanism during deformation is essential for the interpretation of the mechanical properties, highlighting the roles of the grain size, pre-existing heterogeneous dislocations and retained deformation twins.

## 2. Experimental Material and Procedures

An equiatomic $Co_{20}Cr_{20}Fe_{20}Ni_{20}Mn_{20}$ (in at.%) HEA was prepared via the arc-melting method using pure metals. The alloy was remelted at least four times to promote chemical homogeneity. Hot rolling followed by homogenization at 1000 °C for 1080 min in an Ar-controlled atmosphere was conducted to achieve reasonable homogenization, which led to a structure with equiaxed grains and an average grain size of ~200 μm. The homogenized ingot was cold-rolled to achieve a thickness reduction of 60%, followed by post-deformation annealing at 650, 700 and 750 °C, which are below the recrystallization temperature of the alloy (800 °C), for 60 min.

Measurements of the Vickers microhardness, Hv, were taken for all the samples under a load of 100 gf for a dwell time of 10 s, and each point in the values reported for the Hv represents the average of five separate hardness values. Flat dog bone tensile test samples with gauge dimensions of $10.0 \times 2.5 \times 0.8$ mm$^3$ were cut from the annealed sheet using

electro-discharge machining. Tensile tests were conducted at room temperature with a load capacity of 2 kN using an initial strain rate of $1.0 \times 10^{-3}$ s$^{-1}$ with a Santam universal testing machine. Two samples were tested for each condition to achieve good reproducibility. The equipment was adapted for the testing of miniature samples. Stress–strain curves were plotted for each specimen, and the values of the UTS were derived directly from the curves. The elongations were estimated by carefully measuring the gauge lengths before and after tensile testing.

The phase constituents of the annealed samples were determined using X-ray diffraction (XRD), employing Cu K$\alpha$ radiation (wavelength $\lambda$ = 0.154 nm) at 45 kV and a tube current of 200 mA. The XRD measurements were performed over a 2$\theta$ range from 35° to 55° using a scanning speed of 0.6°·min$^{-1}$. To evaluate the microstructures before and after the tensile testing, the specimens were mechanically ground using SiC paper and polished using diamond and colloidal-silica suspensions. The microstructures on the surface (TD plane) of the deformed specimen were evaluated. The polished specimens were analyzed using electron channeling contrast imaging (ECCI) and electron backscatter diffraction (EBSD) with a JEOL-7900 FESEM with a working distance of 18 mm, accelerating voltage of 20 kV and step sizes of 1.7 and 0.4 μm. The grain size measurement was conducted based on the EBSD results. The Oxford energy-dispersed spectroscopy (EDS) detector was used in the same microscopy evaluation for local chemical analysis.

## 3. Results

### 3.1. Microstructure after Post-Deformation Annealing

A set of representative ECCI images with corresponding XRD patterns are shown in Figure 1 for the CoCrFeNiMn HEA processed through medium cold rolling followed by PDA at the temperatures of 650 and 700 °C for 60 min. It is apparent, according to the XRD patterns, that the microstructures consist of a single *fcc* phase with no characteristic peaks of Cr-rich *bcc* or Cr-rich sigma phases. It is important to note that the formation of these phases was reported earlier during annealing at the investigated temperatures after severe or heavy plastic deformation (e.g., >80% cold rolling) [19–22]. The single-phase microstructure represented in Figure 1a contains ultrafine (~0.5 μm) and fine (~5 μm) recrystallized grains with an average grain size of ~2.8 μm and non-recrystallized areas with an average size of ~75 μm, of which the latter are shown by arrows. Annealing twins clearly appeared in the fine recrystallized grain, as shown in Figure 1a. Nevertheless, the non-recrystallized areas reveal typical deformed microstructures containing highly strained areas with complicated non-uniform contrasts due to the presence of high-density defects. The structure clearly shows heterogeneity characteristics such as the range of grain size, pre-existing heterogeneous dislocations, retained deformation twins and annealing twins. A comparison of this microstructure with the microstructure of the alloy after annealing at a higher temperature (Figure 1b) indicates a significant difference at first glance. It seems that a recrystallized microstructure formed with bimodal grain size distributions of ~1.5 and ~6 μm containing annealing twins that appeared in larger grains after PDA at 700 °C. Close inspection of Figure 1b indicates that a very fine deformed area, marked in a circle, survived after annealing. It is important to note that annealing at the higher temperature led to the formation of full recrystallization with an average grain size of ~9 μm (not shown).

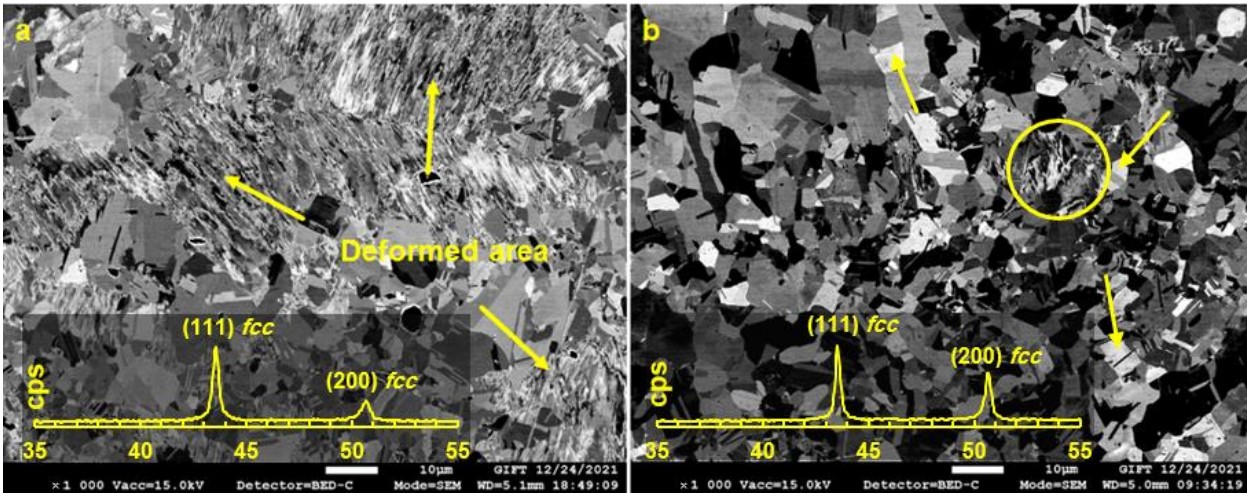

**Figure 1.** ECCI images with corresponding XRD patterns of the CoCrFeNiMn HEA after cold rolling followed by annealing at (**a**) 650 and (**b**) 700 °C for 60 min. Surviving deformed areas after annealing are represented with arrows and in a circle in (**a**,**b**), respectively. Annealing twins are shown by yellow arrows in (**b**).

### 3.2. Mechanical Properties

The measured values of the microhardness after homogenization, cold rolling and PDA at 650, 700 and 750 °C for 60 min are summarized in Table 1. These results indicate that the values of hardness for the homogenized and cold-rolled samples are 145 and 320 Hv, respectively. The hardness decreases constantly with the increase in the annealing temperature to 200 Hv when annealing at 750 °C. It is important to note that there is no difference between the microhardness values of the samples after rolling and PDA at 650 °C. Figure 2a shows representative plots of engineering stress–engineering strain after cold rolling followed by PDA. A summary of the measured values for the yield stress (YS), ultimate tensile strength (UTS), strain-hardening exponent, and uniform and total elongations are given in Table 1. The results noted for the deformed specimens are shown, indicating a high strength with relatively low ductility in comparison with the homogenized condition. This suggests that the annealing of the deformed sample is necessary for improving the ductility. The results clearly show an improved ductility at the expense of decreased strength induced by the increasing annealing temperature. However, the samples annealed at 650 and 700 °C reveal a good combination of strength and ductility, including high UTS values of 1040 and 970 MPa, together with acceptable uniform elongations of 12 and 23%, respectively. Annealing at 650 °C slightly facilitates the restoration of ductility in uniform deformation without decreasing the strength.

**Table 1.** Summary of the mechanical properties of the homogenized, deformed and annealed CoCrFeNiMn HEA.

| Condition | HV$_{0.1}$ | YS (MPa) | UTS (MPa) | Strain-Hardening Exponent | Uniform El. (%) | Total El. (%) |
|---|---|---|---|---|---|---|
| Homogenized | 145 ± 4 | 305 | 600 | 0.33 | 34 | 50 |
| CR | 320 ± 10 | 655 | 1125 | - | 9 | 16 |
| PDA at 650 °C | 320 ± 8 | 650 | 1040 | 0.14 | 12 | 17 |
| PDA at 700 °C | 225 ± 6 | 590 | 970 | 0.18 | 23 | 30 |
| PDA at 750 °C | 200 ± 8 | 540 | 880 | 0.21 | 34 | 41 |

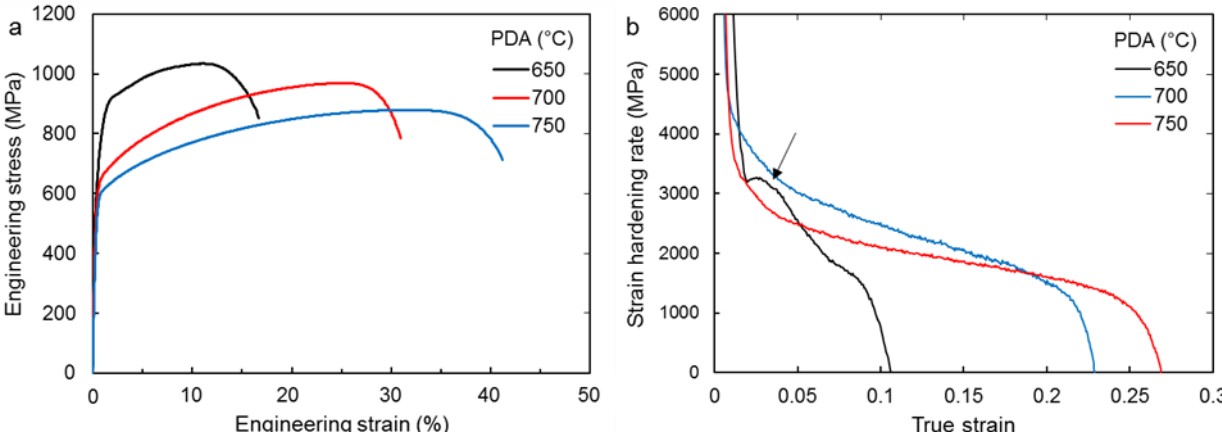

**Figure 2.** (**a**) Stress–strain curves of the CoCrFeNiMn HEA after cold rolling at an initial strain rate of $1.0 \times 10^{-3}$ s$^{-1}$ followed by annealing at 650, 700 and 750 °C for 60 min. (**b**) Strain-hardening rate versus true strain for different conditions. The black arrow in (**b**) represents a slight increase in the hardening rate in stage II.

The flow curve in the uniform deformation region is represented as a power law relation in which the exponent is known as the strain-hardening exponent, and there is a direct correlation between this value and uniform deformation. The results indicate that this value, after PDA, is significantly lower than the strain-hardening exponent of the alloy in the homogenized condition. A higher value of the strain-hardening exponent represents a higher potential of the material for dislocation generation and movement. In fact, there is a correlation between the strain-hardening exponent value and the deformation mechanism. The strain-hardening rate ($d\sigma/d\varepsilon$) versus the true strain curve is plotted in Figure 2b to evaluate the strain-hardening response of the HEA after PDA at different temperatures. The strain-hardening rate curves of the annealed samples at 700 and 750 °C show three stages depending on the slope variation, as reported earlier for *fcc* materials [32–35]. There is a rapid decrease in the strain-hardening rate below the true strain of ~0.04 in stage I, demonstrating an elastic-to-plastic transition. A gradual decrease in the strain-hardening rate is observed after increasing the strain in stage II, which indicates the activation of mechanical twinning during plastic deformation. Finally, a rapid decrease with a further increase in strain occurs in stage III, highlighting dislocation slip as the dominant deformation mechanism. Nevertheless, the strain-hardening rate curve of the annealed sample at 650 °C shows more complicated behavior, including four stages, as detected previously in a HEA with a heterogeneous microstructure [26]. In this case, in stage I, the strain-hardening rate decreases significantly, followed by a slight increase in the hardening rate in stage II (shown by a black arrow). In stage III, the strain-hardening rate decreases again slightly, and then it decreases rapidly to form stage IV.

*3.3. Microstructure of the Annealed Samples after Plastic Deformation*

Figure 3 shows a set of EBSD inverse pole figures (IPFs) in the [001] direction and kernel average misorientation (KAM) maps for the samples annealed at 650 and 700 °C Figure 3a,b before and Figure 3c,d after plastic deformation, respectively, conducted through tensile testing. These images clearly represent the typical microstructure of partial recrystallization, with microstructures of heterogeneous and bimodal grain size distributions after annealing at 650 and 700 °C, respectively. The recrystallized grains and the non-recrystallized areas are distinguishable based on the crystallographic characteristics represented in IPF images and misorientation distributions represented in the KAM maps. The KAM maps represent more sluggish restoration phenomena at lower annealing temperatures. The circles represented in Figure 3b highlight the bimodal grain size distribution in the microstructure. No significant change in grain size is observed in the IPF maps due to the small deformation. However,

the average KAM maps suggest an increase in the KAM average after plastic deformation, which is more effective in the soft recrystallized area shown in Figure 3c.

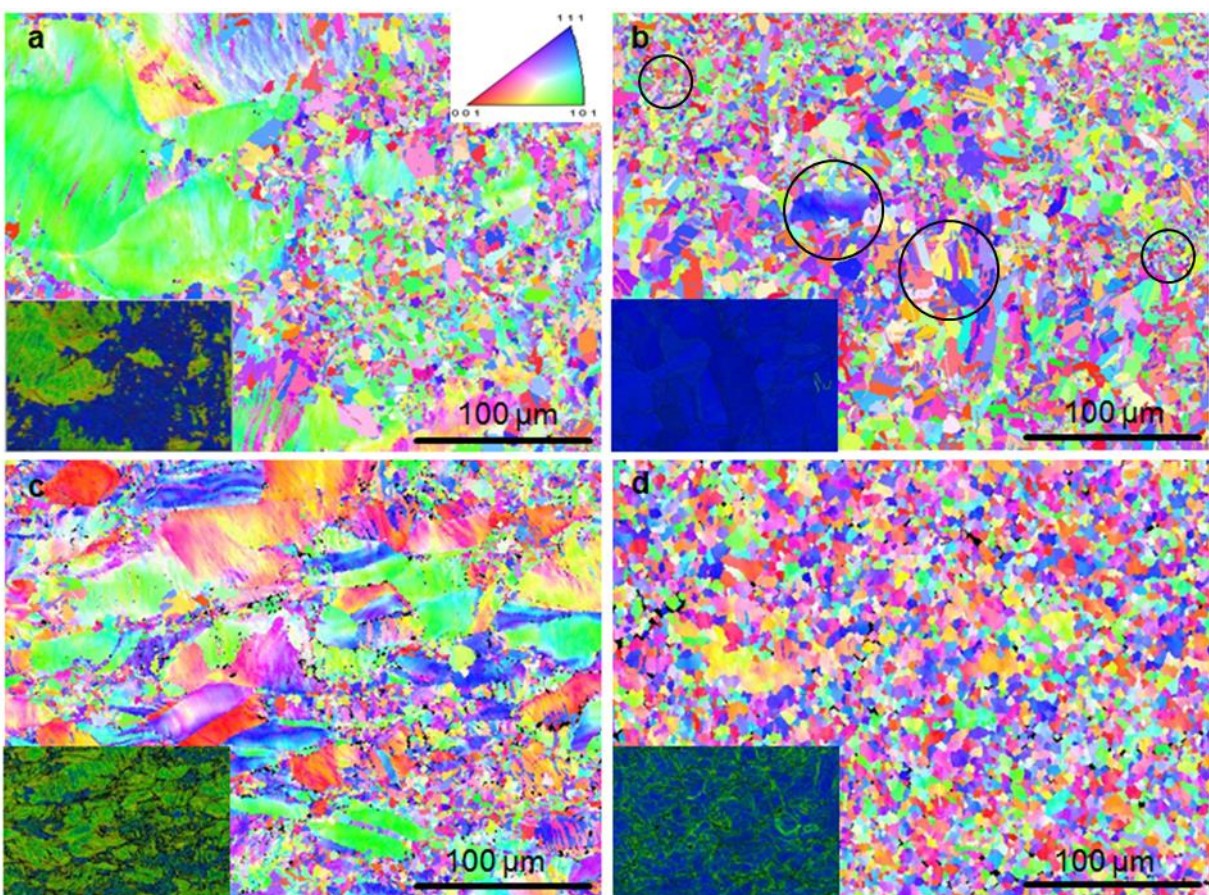

**Figure 3.** EBSD inverse pole figure images together with the KAM of the CoCrFeNiMn HEA after cold rolling followed by annealing at (**a**) 650 and (**b**) 700 °C for 60 min, showing partial recrystallization with microstructures of heterogeneous and bimodal grain size distributions after annealing. Different areas with different grain sizes are shown by circles in (**b**). The images represented in (**c**,**d**) were captured from the deformed microstructures of the annealed samples at 650 and 700 °C, respectively, after tensile testing. The direction of the scale bars was aligned with the tensile direction.

Figure 4 shows more details of the deformed microstructure after annealing at Figure 4a–c 650 and Figure 4d–f 700 °C followed by tensile testing. This figure contains representative ECCI images Figure 4a,d, showing the IPF with the corresponding KAM Figure 4b,e and image quality (IQ) maps Figure 4c,f at a higher magnification compared with Figure 3. IQ is a metric describing the quality of a diffraction pattern. Figure 4a,d illustrates the presence of annealing and deformation twins together with stacking faults, labeled as AT, DT and SF, in the fine grains (>3 μm). Basically, the thickness of the DTs is on the nanoscale and less than that of the ATs. The image in Figure 4a was taken of the recrystallized area; thus, it is reasonable to assume that DTs and SFs formed during plastic deformation. The heterogeneous structures containing ultrafine and fine grains, together with relatively large deformed areas and recrystallized microstructures with bimodal grain size distributions, are shown in Figure 4b,e, respectively. Figure 4c,f reveals the distribution of twins in the microstructure after plastic deformation. A high density of retained deformation twins (RDT) were detected in the deformed areas, together with annealing twins in the recrystallized grains. One of the important aims of this heat treatment is to sustain the DTs because they can act as strong barriers against dislocation movement and improve strength. However, close inspection of Figure 4c,f reveals the formation of

deformation twins (DT) during plastic deformation (marked by arrows). The results suggest that the area fraction of deformation twins increases during plastic deformation, which confirms the theory of the formation of deformation-induced plasticity in the recrystallized grains. The grain size of these grains is higher than the critical grain size for suppressing DT formation during training (~2 μm) [35].

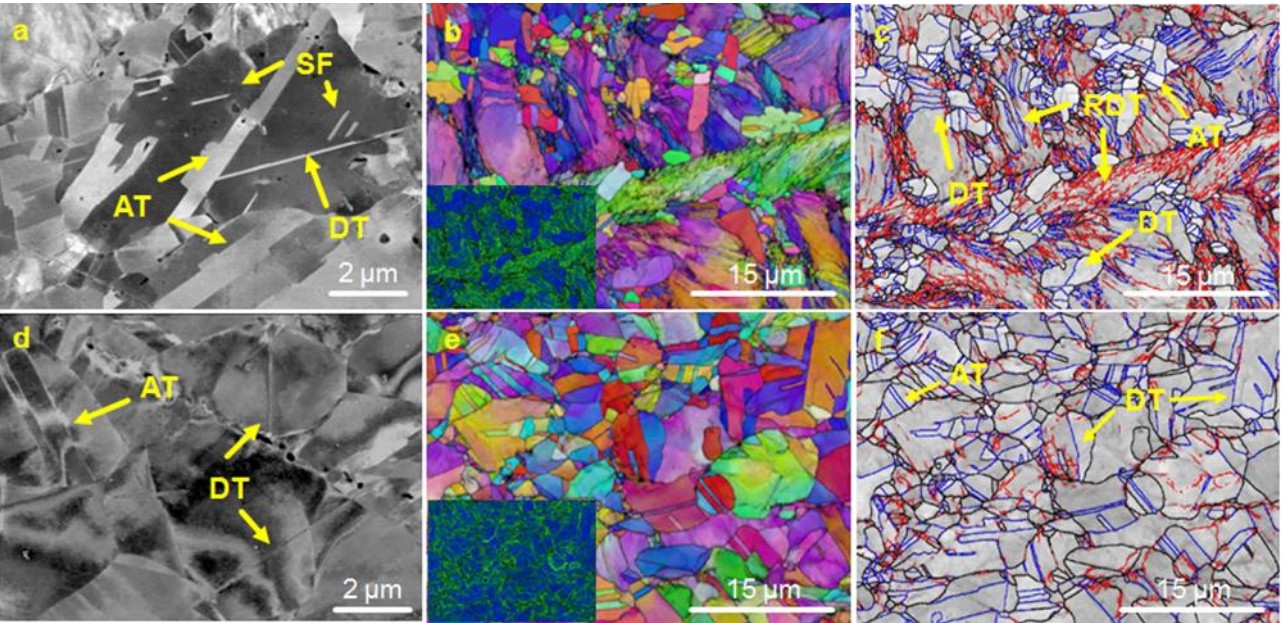

**Figure 4.** Microstructures of samples annealed at (**a**–**c**) 650 and (**d**–**f**) 700 °C after tensile testing at room temperature represented in (**a**,**d**) ECCI images and (**b**,**e**) EBSD inverse pole figures with corresponding KAM and (**c**,**f**) image quality maps. Stacking faults (SF), annealing twins (AT), retained deformation twins (RDT) and deformation twins (DT) are shown by arrows. The image quality maps represent twins' distribution in the microstructure.

## 4. Discussion

### 4.1. Significance of the Heterogeneous Microstructure

Microstructure engineering of the studied HEAs led to an enhanced strength–ductility trade-off due to the formation of a complex heterogeneous structure. This structure was produced using a simple thermomechanical treatment including medium cold rolling followed by PDA at the intermediate temperature of 650 °C for 60 min. It was previously shown that grain refinement alone is not a promising procedure for achieving strength and ductility synergy [26,35,36]. The heterogeneity due to the controlled recrystallization phenomenon in deformed materials which have an inhomogeneous nature of plastic deformation during cold working is a fundamental aspect of the heterogeneous structure [37]. PDA at intermediate temperatures facilitates the formation of a partially recrystallized microstructure and heterogeneous structure. HEAs exhibit relatively high recrystallization and coarsening temperatures and a slow grain-coarsening behavior during PDA due to their inherent sluggish diffusion and high lattice distortion [35,36,38]. This provides a relatively wide temperature range for partial recrystallization, which facilitates easier tuning of the microstructure during PDA.

The heterogeneous structure, with no precipitates such as brittle sigma, was achieved in CoCrFeNiMn processed by cold rolling followed by PDA at 650 °C. It was reported earlier that heavy deformation leads to the formation of this undesirable phase during annealing under conditions similar to those applied in this investigation [21,31]. It seems that applying lower plastic deformation kinetically prevents the formation of these precipitates. In any case, this single-phase alloy with a heterogeneous microstructure exhibits strength–ductility synergy with very high strengths of over ~1 GPa and uniform elongations of over 12%.

The deformed areas contribute to this high strength, and the ultrafine grains provide a good combination of strength and ductility. Annealing at higher temperatures leads to the recrystallization of deformed areas, forming new, fine, equiaxed grains and coarsening the ultrafine/fine recrystallized grains, thus embellishing the bimodal grain size distribution. A microstructure with the characteristic of bimodal grain size distribution is defined as a composite structure, in which the strength and ductility are provided by the fine and coarse grains, respectively. In fact, coarse grains can accommodate more strain due to their potential for the formation of deformation twins. This microstructure provides an excellent combination of strength and ductility, with high strengths of over ~950 MPa and uniform elongations of over 20%. Further increasing the annealing temperature provides a fully recrystallized structure with fine grains with an average grain size of ~9 μm, which significantly restores ductility at the expense of reducing the strength. In other words, the heterogeneous microstructure nearly disappeared after annealing at >700 °C.

### 4.2. Deformation Mechanisms in the Heterogeneous Structure

The formation of complex heterogeneous structures affects deformation mechanisms and changes strain hardenability during straining in a way that leads to the improvement of mechanical properties. Basically, the CoCrFeNiMn alloy with a simple *fcc* crystal structure has great potential for easy deformation via dislocation slip due to the large number of slip systems, together with deformation twinning as a result of its low SFE [18]. These two processes are known as the controlling deformation mechanisms in CoCrFeNiMn HEAs with a homogeneous grain structure. Nevertheless, the heterogeneous structure provides different circumstances. The interactive coupling between the recrystallized area with grain sizes spanning from hundreds of nanometers to several micrometers (0.5 to 5 μm) and the deformed area produces a synergistic effect in which the integrated property exceeds the prediction through the rule of mixtures. The key factor is the formation of geometrically necessary dislocations (GNDs) in the soft recrystallized area, which pile up and accumulate near the area boundaries, therefore producing back stress in the soft areas. It was proposed that the pile-up of GNDs generates back stress in the soft area and causes stress concentration at the interface of the recrystallized and non-recrystallized areas. It promotes the yielding of non-recrystallized areas to induce stress and strain relaxation at the interface. The noted stress concentration generates a stress field in the same direction as the applied stress, called forward-stress, which provides additional strengthening due to the coupled effects of back-stress and forward-stress. These phenomena occur primarily due to the hetero-deformation of soft and hard areas, known as hetero-deformation-induced (HDI) hardening [24]. In fact, this process is activated as an additional strengthening mechanism, together with well-known deformation mechanisms in HEAs, including solid solution strengthening, dislocation hardening and grain boundary strengthening.

The strain rate hardening plot represented in Figure 2b provides a useful way to evaluate the deformation mechanism. The strain rate hardening behavior with an extended plateau in stage II suggests a deformation mechanism due to mechanical twinning. Accordingly, it is expected that the portion of deformation twinning as a deformation mechanism in the annealed sample at 750 °C is higher than that observed in other conditions. Extraordinary strain hardening behavior was observed in the case of the sample annealed at 650 °C, which showed an increasing strain-hardening rate in stage II. It was suggested that the heterogeneous structure plays an important role in this extraordinary behavior [26,39]. In the initial stage of tensile loading, the strain-hardening rate of the fine-grained material drops in a steeper manner than that of the coarse-grained material due to the lack of mobile dislocations. However, after further straining and upon yielding, the dislocations quickly multiply in the fine grains to accommodate the applied constant strain rate, and GNDs form due to the heterogeneous microstructure, with both phenomena leading to the increase in the strain-hardening rate [40]. The activation of slip and deformation twins with a further increase in the strain are responsible for decreasing the strain-hardening rate in the next stages.

The results presented in Figure 4 suggest that DTs and SFs are dynamically generated in recrystallized grains with an average grain size larger than 3 µm during tensile deformation. The KAM maps indicate strain/stress partitioning at the interface between the soft areas (micron-size grains) and hard (deformed) areas and, therefore, the formation of a high stress concentration at the interface during tensile deformation. The formation of DTs and SFs is known as important deformation mechanism during plastic deformation at low and room temperatures in alloys with low SFE which can be retained to some extent after partial recrystallization. It is well-established that the retained DTs and SFs can also support strength enhancement and the strain-hardening ability [41]. The formation of GNDs at the interface of the recrystallized and non-recrystallized areas may be responsible for increasing the KAM after plastic deformation. The high stress at the boundaries may facilitate the driving force of deformation twinning in larger grains induced by the emission of partial dislocations. It appears that the fraction of these twins increases with increasing plastic strain (Figure 4).

### 4.3. Strengthening Mechanisms and the Role of Hetero-Deformation Induced Hardening

Several strengthening mechanisms, including pre-existing dislocation strengthening, retained deformation twins, grain refinement and HDI, can be regarded as the main causes of the extraordinary mechanical properties of the studied material. The results indicate that the coupling effect of HDI strengthening and strain hardening, combined with the formation of DTs and SFs, contributes to the excellent strength–ductility synergy of the samples annealed at 650 and 700 °C. The pre-existing dislocations in the deformed area play an important role in enhancing the mechanical properties. They have two important roles, including enhancing the yield strength by providing strong obstacles to dislocation motion and acting as sources for the emission of Shockley partial dislocations at a higher stress to form DTs and SFs during plastic deformation [42]. The strain-hardening exponent of the alloy after PDA is significantly lower compared with that in the homogenized condition, which leads to a lower uniform elongation. It seems that the deformed area containing pre-existing dislocations or very fine grains is responsible for decreasing the strain-hardening ability of the alloy. The strain-hardening ability is higher when the average grain size is higher than 2 µm due to the easier movement and generation of dislocations, together with deformation twinning, which facilitate higher uniform deformation. A lower chance of deformation twinning for very small grain sizes (<2 µm) during straining led to lower strain hardening in the CoCrFeNiMn HEA [35].

Figure 5 shows the yield stress plot in terms of grain size (Hall–Petch relationship) in different ranges of nano, ultrafine, fine and coarse grains. It demonstrates that there is a marked contrast between the yield stress calculated using the Hall–Petch relationship and the experimental test of the sample undergoing heat treatment at 650 °C due to HDI hardening in this sample. The graph reveals that the mechanical properties of the HEA after severe plastic deformation are lower than the Hall–Petch curve. Thus, applying severe or heavy plastic deformation to the CoCrFeNiMn HEA is not a useful approach to overcoming the strength–ductility trade-off. Nevertheless, the results of the present investigation show that the mechanical properties can be significantly improved by creating a heterogenous microstructure containing a deformed area and fine recrystallized grains, in addition to a bimodal grain structure, in the CoCrFeNiMn HEA.

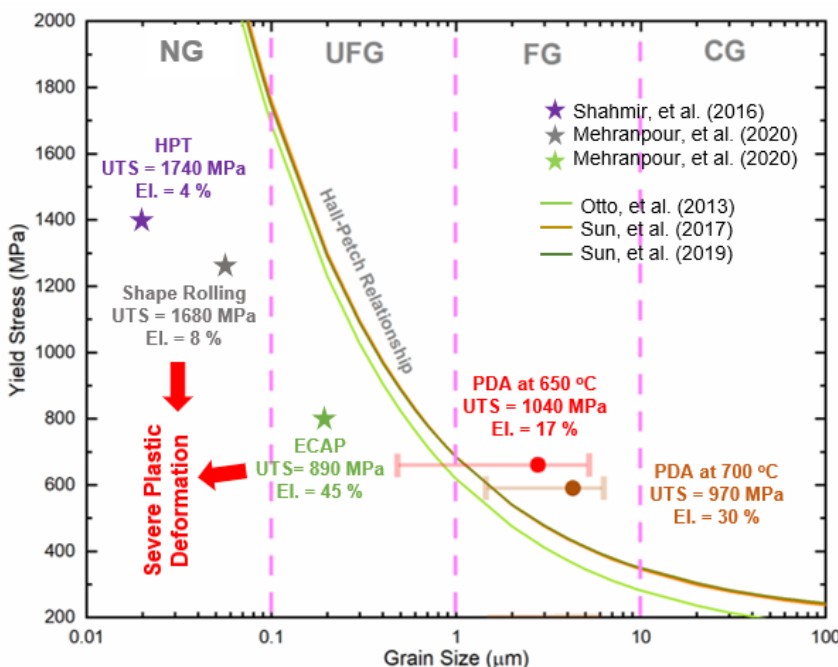

**Figure 5.** Yield stress versus grain size for the CoCrFeNiMn HEA after different thermomechanical processes [6,19,21,22,43,44]. HPT: High-pressure torsion, ECAP: Equal-channel angular pressing, PDA: Post-deformation annealing, UTS: Ultimate-tensile stress, El: Elongation to failure.

## 5. Conclusions

The main approach of the present investigation was directed specifically towards microstructure tailoring of the CoCrFeNiMn HEA by creating a heterogeneous microstructure to achieve a good combination of strength and ductility. A CoCrFeNiMn HEA was processed via medium plastic deformation through cold rolling with a 60% area reduction followed by post-deformation annealing at 650, 700 and 750 °C for 60 min. The noted procedures provided a wide range of features, from heterogeneous and bimodal grain structures to homogeneous structures with fine grains. Annealing at 650 °C led to the formation of non-recrystallized areas with an average size of ~75 μm, together with ultrafine (~0.5 μm) and fine (~5 μm) recrystallized grains with an average grain size of ~2.8 μm. Increasing the annealing temperature up to 700 °C facilitated the formation of bimodal grain size distributions of ~1.5 and ~6 μm, containing annealing twins that appeared in the larger grains. The heterogeneous structure was replaced with a fully recrystallized structure with an average grain size of ~9 μm after annealing at higher temperatures. The results indicate that the coupling effect of hetero-deformation-induced strengthening and strain hardening, combined with the formation of deformation twins and stacking faults, contributes to the excellent strength–ductility synergy of samples annealed at 650 and 700 °C.

**Author Contributions:** Conceptualization, H.S.; methodology, P.S. and S.A.A.S.; validation, H.S., M.S.M. and S.A.A.S.; formal analysis, P.S. and S.A.A.S.; investigation, P.S. and S.A.A.S.; resources, H.S. and C.S.L.; data curation, H.S. and P.S.; writing—original draft preparation, H.S.; writing—review and editing, M.S.M. and S.A.A.S.; visualization, P.S. and S.A.A.S.; supervision, H.S. and C.S.L.; project administration, H.S.; funding acquisition, H.S. All authors have read and agreed to the published version of the manuscript.

**Funding:** This research received no external funding.

**Data Availability Statement:** Data is unavailable due to privacy or ethical restrictions.

**Conflicts of Interest:** The authors declare no conflict of interest.

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
