# Peer review of "Hetero-Deformation Induced Hardening in a CoCrFeNiMn High-Entropy Alloy"

_crystals, doi:10.3390/cryst13050844_

Round 1

Reviewer 1 Report

referee report 

crystals-2389784-peer-review-v1

Hetero-Deformation Induced Hardening in a CoCrFeNiMn High-Entropy Alloy

Hamed Shahmir et al.

This manuscript reports on the fabrication and characterization of a CoCrFeNiMn high-entropy alloy pro-13 cessed by cold rolling 

followed by post-deformation annealing over a range of temperatures which led to a wide range of grain sizes. The resulting 

microstructures were thoroughly characterized employing SEM, EDS and EBSD. The topic is well suited for Crystals.

The manuscript comprises 5 figures, 1 table, and 45 references are provided by the authors.

The manuscript is well organized, well written in proper English. All the figures included are well prepared except the point listed 

below.

There are some points which require attention prior to publication:

# 166: Table 1: Do not use a linebreak within an element within a chemical formula

# Please give the maker of the EBSD system including some details like working distance, voltage, etc.

# In which direction is the IPF mapping?

# Please give some more details on the KAM mapping. Which amount of neighbors you have included?

# For the KAM mappings, the corresponding color code (degrees) is missing. Please provide it (one is enough for all images in the 

   manuscript).

# Please explain image quality to all non-experienced people.

# What are the various colors for the GBs indicated?

# All captions of Figs. 3 and 4 need to describe all details visible in the images.

Overall, the present manuscript provides interesting data material, but there are some points which require attention prior

to publication.

see above

Author Response

We thank the reviewer for providing the comments on our manuscript.  Our responses are given below in bold font and changes are marked in the manuscript using yellow highlight.

Reviewer 1

This manuscript reports on the fabrication and characterization of a CoCrFeNiMn high-entropy alloy processed by cold rolling followed by post-deformation annealing over a range of temperatures which led to a wide range of grain sizes. The resulting microstructures were thoroughly characterized employing SEM, EDS and EBSD. The topic is well suited for Crystals.

The manuscript comprises 5 figures, 1 table, and 45 references are provided by the authors. The manuscript is well organized, well written in proper English. All the figures included are well prepared except the point listed below.

There are some points which require attention prior to publication:

 # 166: Table 1: Do not use a linebreak within an element within a chemical formula.

There is no linebreak in the original uploaded file. It seems to me that it has happened during making PDF by the system.

# Please give the maker of the EBSD system including some details like working distance, voltage, etc.

Some information was added and highlighted in the manuscript.

# In which direction is the IPF mapping?

The direction is [001]. It was added to the manuscript.

# Please give some more details on the KAM mapping. Which amount of neighbors you have included?

The number of neighbors in KAM is 5.

# For the KAM mappings, the corresponding color code (degrees) is missing. Please provide it (one is enough for all images in the manuscript).

It was added to the manuscript.

# Please explain image quality to all non-experienced people.

It was added to the manuscript.

# What are the various colors for the GBs indicated?

All GBs are shown by black color in IQ maps. Different colors depict distinguished twins.

# All captions of Figs. 3 and 4 need to describe all details visible in the images.

Some explanations were added.

Reviewer 2 Report

The paper is interesting and well written; the text is clear and easy to read, and the results are adequately discussed. This makes the paper possible for publishing in Crystals.

However, to accept, the paper should be subject to some revision, and additionally the article needs to be supplemented with some results. The detailed remarks are as follows:

1.     Abstract: provide some numerical values as properties. It looks general.

2.     Introduction: Avoid blocks of references, for example: ‘[...] has been drawn attention due to its high ductility even at cryogenic temperatures, fracture toughness, phase stability and corrosion resistance [1-5].. ‘[...] such as grain refinement using severe plastic deformation, thermomechanical treatment and precipitation hardening [8-16].’…’[…] to achieve optimum mechanical properties [18-23]…[…] HEAs which may offer superior mechanical properties compared with the fully recrystallized material [27-31]. As these do not emphasize the particular aspects of the cited papers. In particular, when citations are made about specific technical aspects, single / double references, e.g. [1, 2] are encouraged. It is strongly suggested that the references make in-depth comments on the content of the cited papers.

3.     Results: The paper lacks a summary/comparative XRD diagram for all the variants investigated (after the rolling process and after heat treatment under different conditions). It would be good to complete the article, in the paper only fragments of the XRD diagram for the two variants are presented in Fig. 1. Besides, the analysis is presented for a small angle range: 35-55 degrees.

The authors write that they obtained a bimodal microstructure and gave the grain size. It seems necessary to include in the paper the detailed results of the grain size measurement (based on the EBSD results) and present them in the form of a graph, then it will really be seen whether the microstructure is bimodal.

Besides, it would be good to present the results of the disorientation measurement, with ultrafine microstructures this is a very important parameter; this is also the result obtained from the EBSD tests.

Author Response

We thank the reviewer for providing the comments on our manuscript.  Our responses are given below in bold font and changes are marked in the manuscript using yellow highlight.

Reviewer 2

The paper is interesting and well written; the text is clear and easy to read, and the results are adequately discussed. This makes the paper possible for publishing in Crystals.

However, to accept, the paper should be subject to some revision, and additionally the article needs to be supplemented with some results. The detailed remarks are as follows:

  1. Abstract: provide some numerical values as properties. It looks general.

It was modified according to the comment.

  1. Introduction: Avoid blocks of references, for example: ‘[...] has been drawn attention due to its high ductility even at cryogenic temperatures, fracture toughness, phase stability and corrosion resistance [1-5].. ‘[...] such as grain refinement using severe plastic deformation, thermomechanical treatment and precipitation hardening [8-16].’…’[…] to achieve optimum mechanical properties [18-23]…[…] HEAs which may offer superior mechanical properties compared with the fully recrystallized material [27-31]. As these do not emphasize the particular aspects of the cited papers. In particular, when citations are made about specific technical aspects, single / double references, e.g. [1, 2] are encouraged. It is strongly suggested that the references make in-depth comments on the content of the cited papers.

We tried to do that but, in some cases, we need to refer to couple of papers which is common. For example, in the case of “grain refinement using severe plastic deformation, thermomechanical treatment and precipitation hardening [8-16]”, Refs. 8 and 9 cover all three methods of severe plastic deformation, thermomechanical treatment and precipitation hardening. Or in another case: “Most investigations earlier were focused on fully recrystallization and tailoring the microstructure by grain refinement together with appropriate precipitates to achieve optimum mechanical properties [18-23]”, it is not possible to split the references.

  1. Results: The paper lacks a summary/comparative XRD diagram for all the variants investigated (after the rolling process and after heat treatment under different conditions). It would be good to complete the article, in the paper only fragments of the XRD diagram for the two variants are presented in Fig. 1. Besides, the analysis is presented for a small angle range: 35-55 degrees.

There are many investigations including our previous works that reported XRD patterns in cold rolled and initial homogenized conditions and it was shown that characteristic peaks of probable precipitates are appeared in the range of 35-55 degrees [9, 14, 19, 21, 22, 31, 35]. So, in this investigation, we only focused on the annealed samples and it was important to show that the material was single phase in the investigated condition.

 The authors write that they obtained a bimodal microstructure and gave the grain size. It seems necessary to include in the paper the detailed results of the grain size measurement (based on the EBSD results) and present them in the form of a graph, then it will really be seen whether the microstructure is bimodal.

Besides, it would be good to present the results of the disorientation measurement, with ultrafine microstructures this is a very important parameter; this is also the result obtained from the EBSD tests.

The grain sizes were measured based on EBSD results and using related software. This explanation was added to the manuscript. We believe that the presented set of images and other data sufficiently express the microstructures.  The corresponding color code (degrees) for the KAM mappings was added to Fig. 3 and 4 to better presentation of the disorientation in the microstructure.

Reviewer 3 Report

The manuscript presents the original results of a study of the effect of heterogeneous structure and complex microstructures of a high-entropy CoCrFeNiMn alloy (an alloy known as the Cantor alloy) after cold rolling followed by post-deformation annealing at temperatures ranging from 650 °C, 700 °C, 750 °C. The results of studies of the microhardness and parameters of the mechanical behavior of the CoCrFeNiMn alloy under quasi-static tension in states with grain sizes from 0.5 µm to 200 µm, including alloy with bimodal grain distributions, are presented.

It is shown that the creation of a heterogeneous microstructure in the CoCrFeNiMn HEA by using thermomechanical treatment makes it possible to improve the mechanical properties of the Cantor’s alloy and obtain a better combination of strength and ductility compared to cast samples. The considered processing modes made it possible to obtain the strength ~ 1 GPa and a uniform elongation of more than 12% for the Cantor’s alloy.

An important result of the study is the establishment of the possibility of achieving a synergistic effect of strength and ductility of an annealed sample of Cantor’s alloy at 650 and 700 °C as a result of the coupling effect of hetero-deformation-induced strengthening and strain hardening in combination with the formation of deformation twins and stacking faults.

The manuscript is well structured. The manuscript has good quality illustrations. The reference list is satisfactory.

The results presented in the article are new and may be of interest to a wide range of specialists, graduate students and students who study the structure and properties of high-entropy alloys.

The manuscript needs some minor additions.

Notes:

1) Specify the specific brand of Santam universal testing machine and the extensometers and force sensors used.

The information is important for evaluating the accuracy of determining forces and displacements when testing mini samples of HEA. The tested flat dog bone shape mini samples with gauge dimensions of 10.0×2.5×0.8 mm3 were did not meet the requirements of the ASTM E8M standard. Please note that universal testing machines are designed to test samples in accordance with current standards.

2) The used heat treatment mode leads to the formation of grains with a size of ~200 μm (Line 92). The thickness of the samples used for testing is 0.8 mm; 800 µm (Line 99). Thus, only 4 layers of grains can be located along the thickness of the samples. The question arises about the influence of the anisotropy of grain properties on the results of evaluating the deformation of mini-samples, which were used to determine the mechanical characteristics presented in Table 1 (Line 166) and Figure 2 (Line 190).

When testing samples with grain sizes of 0.5, 5, 75 μm (Lines 126-128), this issue does not arise.

3) The manuscript should be supplemented with an explanation of which samples were used for microhardness measurements (Line 96). Measurements of microhardness on ingots and on the annealed sheet used for electro erosion manufacturing of specimens can be differing.

4) The text (Line 210) has to complete with the time in min (“(b) 700 °C for 60 (?)”).

 The text (Line 210) has to complete with the time in min (“(b) 700 °C for 60 (?)”).

Author Response

We thank the reviewer for providing the comments on our manuscript.  Our responses are given below in bold font and changes are marked in the manuscript using yellow highlight.

Reviewer 3

The manuscript presents the original results of a study of the effect of heterogeneous structure and complex microstructures of a high-entropy CoCrFeNiMn alloy (an alloy known as the Cantor alloy) after cold rolling followed by post-deformation annealing at temperatures ranging from 650 °C, 700 °C, 750 °C. The results of studies of the microhardness and parameters of the mechanical behavior of the CoCrFeNiMn alloy under quasi-static tension in states with grain sizes from 0.5 µm to 200 µm, including alloy with bimodal grain distributions, are presented.

It is shown that the creation of a heterogeneous microstructure in the CoCrFeNiMn HEA by using thermomechanical treatment makes it possible to improve the mechanical properties of the Cantor’s alloy and obtain a better combination of strength and ductility compared to cast samples. The considered processing modes made it possible to obtain the strength ~ 1 GPa and a uniform elongation of more than 12% for the Cantor’s alloy.

An important result of the study is the establishment of the possibility of achieving a synergistic effect of strength and ductility of an annealed sample of Cantor’s alloy at 650 and 700 °C as a result of the coupling effect of hetero-deformation-induced strengthening and strain hardening in combination with the formation of deformation twins and stacking faults.

The manuscript is well structured. The manuscript has good quality illustrations. The reference list is satisfactory.

The results presented in the article are new and may be of interest to a wide range of specialists, graduate students and students who study the structure and properties of high-entropy alloys.

The manuscript needs some minor additions.

Notes:

1) Specify the specific brand of Santam universal testing machine and the extensometers and force sensors used.

The information is important for evaluating the accuracy of determining forces and displacements when testing mini samples of HEA. The tested flat dog bone shape mini samples with gauge dimensions of 10.0×2.5×0.8 mm3 were did not meet the requirements of the ASTM E8M standard. Please note that universal testing machines are designed to test samples in accordance with current standards.

Some information was added to the manuscript. It is important to note that we used similar equipment which is an Iranian brand in our previous investigations and the equipment was adapted for testing miniature samples. As noted in the manuscript, the elongations were estimated by carefully measuring the gauge lengths before and after tensile testing which is the conventional method for the miniature samples.

2) The used heat treatment mode leads to the formation of grains with a size of ~200 μm (Line 92). The thickness of the samples used for testing is 0.8 mm; 800 µm (Line 99). Thus, only 4 layers of grains can be located along the thickness of the samples. The question arises about the influence of the anisotropy of grain properties on the results of evaluating the deformation of mini-samples, which were used to determine the mechanical characteristics presented in Table 1 (Line 166) and Figure 2 (Line 190).

When testing samples with grain sizes of 0.5, 5, 75 μm (Lines 126-128), this issue does not arise.

Figure 2 represents the mechanical properties of the alloys after thermomechanical treatment with fine and ultrafine grains. So, the noted point is not relevant for these conditions. It may affect the mechanical properties of the homogenized condition. It is important to note that the similar results reported earlier for the homogenized alloy using standard tensile tests.

3) The manuscript should be supplemented with an explanation of which samples were used for microhardness measurements (Line 96). Measurements of microhardness on ingots and on the annealed sheet used for electro erosion manufacturing of specimens can be differing.

The required explanation was added to the manuscript.

4) The text (Line 210) has to complete with the time in min (“(b) 700 °C for 60 (?)”).

It was corrected.

Comments on the Quality of English Language

The text (Line 210) has to complete with the time in min (“(b) 700 °C for 60 (?)”).

It was corrected.

Round 2

Reviewer 2 Report

Thank you very much for partially addressing my comments. I cannot agree with the explanation of the 'Introduction'. It is written in very general terms in places, and one might be tempted to comment more deeply in some places.

How certain is it that the grain size was determined, not the subgrain size? Why did the authors decide not to present graphically, in the form of graphs, the curves showing grain size and the proportion of large and small disorientation angles. The representation in the photo alone is insufficient. For me, there is still no confirmation that the microstructure is bimodal.

Regarding the XRD results, it is the reader who does not necessarily have the desire and time to look for the results in other works of the authors. The work should be comprehensive.

Author Response

Thank you very much for partially addressing my comments. I cannot agree with the explanation of the 'Introduction'. It is written in very general terms in places, and one might be tempted to comment more deeply in some places.

Response: Thank you for the comments. I believe that the Introduction sufficiently covers the necessities for the manuscript’s subject.

How certain is it that the grain size was determined, not the subgrain size? Why did the authors decide not to present graphically, in the form of graphs, the curves showing grain size and the proportion of large and small disorientation angles. The representation in the photo alone is insufficient. For me, there is still no confirmation that the microstructure is bimodal.

Response: EBSD is a powerful equipment to study the microstructure in which high-angle grain boundaries can be distinguished from low angle grain boundaries (as shown in the images represented in the manuscript). In addition, I believe that the EBSD IPF together with IQ and ECCI images represented in Figs. 1, 3 and 4 clearly show that the microstructure is heterogenous.

Regarding the XRD results, it is the reader who does not necessarily have the desire and time to look for the results in other works of the authors. The work should be comprehensive.

Response: I believe that the sufficient information was reported in the manuscript. The investigated material is a well-studied HEA alloy and I think it is not necessary to repeat some information (XRD of the initial and as-rolled or annealed samples at >800 deg C) which have been reported several times in many researches (not only the previous results of the authors).